# Recycling the Purpose of Old Drugs to Treat Ovarian Cancer

**DOI:** 10.3390/ijms21207768

**Published:** 2020-10-20

**Authors:** Mariana Nunes, Miguel Henriques Abreu, Carla Bartosch, Sara Ricardo

**Affiliations:** 1Differentiation and Cancer Group, Institute for Research and Innovation in Health (i3S) of the University of Porto/Institute of Molecular Pathology and Immunology of the University of Porto (Ipatimup), 4200-135 Porto, Portugal; marianaoliveiranunes1992@gmail.com; 2Porto Comprehensive Cancer Center (PCCC), 4200-162 Porto, Portugal; p_abreu@sapo.pt (M.H.A.); carlabartosch@yahoo.com (C.B.); 3Department of Medical Oncology, Portuguese Oncology Institute of Porto (IPOP), 4200-162 Porto, Portugal; 4Department of Pathology, Portuguese Oncology Institute of Porto (IPOP), 4200-162 Porto, Portugal; 5Cancer Biology & Epigenetics Group, Research Center—Portuguese Oncology Institute of Porto (CI-IPOP), 4200-162 Porto, Portugal; 6Faculty of Medicine, University of Porto, 4200-319 Porto, Portugal; 7Department of Sciences, University Institute of Health Sciences (IUCS), CESPU, CRL, 4585-116 Gandra, Portugal

**Keywords:** ovarian cancer, chemoresistance, drug repurposing, ex vivo cultures

## Abstract

The main challenge in ovarian cancer treatment is the management of recurrences. Facing this scenario, therapy selection is based on multiple factors to define the best treatment sequence. Target therapies, such as bevacizumab and polymerase (PARP) inhibitors, improved patient survival. However, despite their achievements, ovarian cancer survival remains poor; these therapeutic options are highly costly and can be associated with potential side effects. Recently, it has been shown that the combination of repurposed, conventional, chemotherapeutic drugs could be an alternative, presenting good patient outcomes with few side effects and low costs for healthcare institutions. The main aim of this review is to strengthen the importance of repurposed drugs as therapeutic alternatives, and to propose an in vitro model to assess the therapeutic value. Herein, we compiled the current knowledge on the most promising non-oncological drugs for ovarian cancer treatment, focusing on statins, metformin, bisphosphonates, ivermectin, itraconazole, and ritonavir. We discuss the primary drug use, anticancer mechanisms, and applicability in ovarian cancer. Finally, we propose the use of these therapies to perform drug efficacy tests in ovarian cancer ex vivo cultures. This personalized testing approach could be crucial to validate the existing evidences supporting the use of repurposed drugs for ovarian cancer treatment.

## 1. Introduction

Ovarian cancer (OC) is the leading cause of gynaecological cancer-related mortality worldwide [1,2]. In Europe, 67,800 women are diagnosed with OC and 44,600 die from this malignancy annually [3]. The most prevalent subtype, high-grade serous carcinoma (HGSC), is particularly lethal, since it develops rapidly and often presents with advanced stage disease [4,5,6]. Treatment options are limited, and typically involve cytoreductive surgery and platinum-based chemotherapy [4,5,6,7]. While many patients initially respond well, approximately 75% of women develop incurable recurrences, contributing to poor survival rates that have not changed substantially for the past years [8]. The main challenge in OC is to unveil a therapeutic strategy to overcome chemoresistance. Many targeted therapies have been approved to treat OC, e.g., poly (ADP-ribose) polymerase (PARP), and vascular endothelial growth factor (VEGF) inhibitors; however, the median progression-free survival (PFS) remains poor [9,10,11]. Even in platinum sensitive relapse and considering only the best responders, the addition of bevacizumab or PARP inhibitors (iPARPs) correlates with a PFS no longer than 21 months [12,13,14,15,16]. Here, we propose an alternative approach based on the use of non-oncological drugs for OC treatment. This concept, called drug repurposing, is based on the knowledge of pharmacokinetics, pharmacodynamics, target identification, bioavailability, toxicity profiles, recommended dosage schemes, and consistent recognition of adverse effects, meaning that oncological indication development can begin at phase II of the clinical trials, making the research process less time-consuming and less expensive [17,18,19,20]. The incorporation of non-oncological drugs for cancer treatment is usually combined with chemotherapeutic agents, target therapies, or other repurposed drugs [21].

In the OC setting, many non-oncological drugs have promising in vitro results and some of them already being tested in clinical trials. These therapeutic compounds include antifungal (itraconazole), antilipidemic (statins), antidiabetic (metformin), antiviral (ritonavir), antiparasitary (ivermectin) drugs, and osteoporosis treatment (bisphosphonates). In this review, we gather information about the most promising drug repurposing strategies for OC treatment and propose a strategy to test these therapeutic options in patient-derived samples.

## 2. Drug Repurposing for Ovarian Cancer Therapy

### 2.1. Statins

Statins are inhibitors of 3-hydroxy-3-methylglutaryl-coenzyme A reductase (HMGCR), located upstream in the mevalonate pathway, which is responsible for cholesterol biosynthesis [22,23,24]. The main clinical use of statins is to reduce plasma cholesterol levels, being a key treatment in the prevention of cardiovascular diseases [25,26,27,28,29]. Statins can also reduce inflammation, influence vascular expansion and remodelling, and inhibit coagulation and fibrinolysis [30,31,32].

The inhibition of HMGCR leads to a cascade of inhibition in downstream proteins (i.e., geranyl pyrophosphate, farnesyl pyrophosphate, and geranylgeranyl pyrophosphate), that play important roles in several signalling pathways, such as membrane trafficking, cell motility, proliferation, differentiation, and cytoskeletal organization [33]. HMGCR is considered a metabolic oncogene as it promotes tumour growth and co-operates with rat sarcoma virus (RAS) to transform cells in colony forming assays [34]. Retrospective studies suggest that statins exhibit an anti-tumoral effect and are associated with a decreased risk of recurrence in several neoplasms, including breast and ovarian cancers [26,35,36,37,38,39,40,41,42,43,44].

In OC setting, it has been demonstrated that statins reduce cell proliferation and migration in vitro leading to a delay in tumour formation and suppress the metastatic capacity in vivo [45,46,47]. Several others experimental studies, using OC cell lines and animal models, demonstrate that statins block the cholesterol biosynthesis leading to apoptosis and tumour cell differentiation, proliferation, invasion, migration, angiogenesis and metastasis, through the activation of multiple signalling pathways [35,36,46,48,49,50,51,52]. Martirosyan et al., demonstrated that lovastatin triggers apoptosis of OC cells as a single agent by blocking HMGCR activity (mevalonate-dependent mechanism) and sensitizing chemoresistant cells to doxorubicin by blocking drug efflux pumps (mevalonate-independent mechanism) (Figure 1) [35]. A recent study performed by Dr Richardson’s group, found that pitavastatin, a lipophilic statin with a long half-life, was able to inhibit the growth of two-dimensional (2D) and three-dimensional (3D) OC cell cultures with a resistant profile to carboplatin, suggesting that pitavastatin has a great potential to treat chemoresistant tumours [53]. Additionally, this study showed that high expression of wild type and gain-of-function tumour protein 53 (*TP53*) variants led to an increased HMGCR expression [53]. This observation is particularly important in HGSC, where dysregulation of *TP53* is an almost ubiquitous event [54,55,56], suggesting that a significant proportion of OC patients could benefit from pitavastatin treatment [53,57].

Regarding OC survival studies, the overall survival (OS) rate for post diagnostic patients is significantly higher in statins users (hazard ratio (HR)) = 0.63; 95% confidence interval (CI) = 0.54–0.74 [58]; HR = 0.87; 95% CI = 0.80–0.95 [59]) compared to non-users. Other studies show that statin users present all-cause mortality and cancer-related mortality rates significantly lower (HR = 0.72; 95% CI = 0.56–0.93 [60]; HR = 0.85; 95% CI = 0.82–0.87 [61]) compared to non-users. A recent study demonstrated that patients who started a statin treatment after being diagnosed with OC had a significant reduced risk of all mortality (adjusted HR = 0.81; 95% CI = 0.72–0.90) compared to non-users, being associated with improved survival [62]. Several other studies show evidences for a significant reduction in OC risk among statin users compared to non-users [26,51,62,63]. In fact, for 13 types of cancers, it was observed a decrease in cancer-related death risk in statin users [61]. The anti-tumoral effects of statins has also been described in variety of other cancer origins, for example, endometrial cancer (HR = 0.83; 95% CI = 0.69–1.01) [58], breast cancer (HR = 0.81; 95% CI = 0.68–0.96) [64], colorectal cancer (fully adjusted HR = 0.71; 95% CI = 0.61–0.84) [65] and prostate cancer (HR = 0.76; 95% CI = 0.66–0.88) [66]. These studies show the effect of statins to prevent the onset of cancer.

Several experimental and observational studies suggest that statins could exhibit anticancer properties or being used as adjuvants to the current OC therapeutic options [67]. In concordance, ongoing clinical trials are testing the effects of different statins in the treatment of OC patients (Appendix A). Statins efficacy has been studied for as cancer treatment as monotherapy or combinatorial therapy, being the last option the most effective [47,68,69,70,71,72].

Statins are one of the most commonly used drugs worldwide to treat cardiovascular diseases. The results on its putative use in OC prevention/recurrence and in a chemoresistant setting are promising but there is much to improve regarding the type of statins, its half-life and dosage.

### 2.2. Metformin

Metformin is used in the treatment of type 2 diabetes since induces anti-hyperglycaemia, mainly through the decrease of insulin resistance and blocking glucose-6-phosphatase that reduces hepatic gluconeogenesis and inhibit gastro-intestinal glucose reabsorption [73,74,75,76]. Metformin inhibit the respiratory-chain complex 1, induce a drop in cellular energy charge, activate glucose uptake through promoting glucose transporter type 4 translocation to the plasma membrane that mediates the activation of liver kinase B1 and adenosine monophosphate-activated protein kinase (AMPK) [75,76,77,78]. Consequently, cellular adenosine triphosphate (ATP) concentrations fall, and the increase in both adenosine diphosphate (ADP)/ATP and adenosine monophosphate (AMP)/ATP ratios triggers AMPK that coordinates a wide range of compensatory, protective, and energy sparing responses, leading to a reduction in hepatic glucose output [75,76,77,78]. Metformin inhibits cell proliferation and insulin signals, blocks protein and fatty acid synthesis, and exhibits anti-inflammatory properties [79,80]. Besides that, it has also been reported that metformin has anti-tumoral properties, since the inhibition of the glucose production will cause alterations in metabolic and endocrine circuits that change some cellular and molecular processes that will influence cancer biology causing oxidation stress and deoxyribonucleic acid (DNA) damage [17,79,81].

Many mechanisms of metformin’s anti-cancer activity have been proposed. Some studies suggest that metformin modulate the immunological and/or anti-inflammatory responses in cancer treatment [82,83]. Other studies demonstrate that metformin inhibit mammalian target of rapamycin (mTOR) through activation of AMPK resulting in reduced cancer cells proliferation [84,85]. Metabolic actions have been proposed related to gluconeogenesis, mitochondrial function, and cellular metabolism [86,87]. Metformin reduces cellular respiration by inhibiting respiratory-chain complex 1 limiting the cancer cell’s metabolic plasticity [73,88,89]. However, cancer cells try to compensate the suppression of oxidative phosphorylation by enhancing glycolysis that is p53-dependent [89]. HGSC normally present absence of functional p53; therefore, cancer cells are incapable to compensate for metformin-induced suppression of oxidative metabolism [90]. In OC, metformin is reported to reverse chemoresistance, avoid epithelial mesenchymal transition (EMT), reduce cancer cell migration, and metastasis [87,91,92,93,94]. In fact, in this tumoral context, several studies demonstrate that metformin blocks cell growth, induces apoptosis, inhibits angiogenesis and metastatic spread, potentiates effectivity of chemotherapeutic agents, and reverses chemoresistance (Figure 1) [77,86,92,95,96,97,98,99,100,101,102,103]. Besides that, metformin, either alone or in combination with cisplatin, inhibit cell viability and angiogenesis, and induces apoptosis in OC cell lines [104]. Another in vitro and in vivo study demonstrated that combining iPARPs and metformin enhances the anti-tumoral effects regardless of breast cancer (BRCA) status [105].

Epidemiological studies reveal that cancer-related mortality rate is significantly lower in metformin patient users and that survival rates is improved in many types of cancer patients, i.e., colorectal, pancreatic, breast, liver and endometrial cancers [106,107]. A recent study demonstrated that metformin significantly prolonged the OS (HR = 0.61; 95% CI = 0.48–0.77) and reduced recurrence risk (HR = 0.50; 95% CI = 0.28–0.92) in endometrial cancer [108]. Regarding endometrial cancer, results showed that metformin amplify the effects of paclitaxel by blocking mTOR-signalling pathway [109]. Other epidemiological studies and meta-analysis demonstrate that metformin reduce OC risk (odd ratio (OD) = 0.61; 95%CI = 0.30–1.25 [110]; OD = 0.57; 95%CI = 0.16–1.99 [111]) and OC-specific mortality [110,111,112,113,114,115,116]. In addition, the 5-years PFS was significantly better for OC patients with type 2 diabetes who were taking metformin (63%) compared to users of other hypoglycaemic drugs (37%) and non-users (23%) (*p* = 0.03) [112]. Corroborating these results, another study demonstrated that 5-years survival rate was significantly better in the group of metformin users (67%) compared to non-users (47%) and that metformin remained an independent predictor of survival (HR = 2.2; 95% CI = 1.2–3.8) [115]. A recent meta-analysis demonstrated that the metformin use in post-diagnostic OC patients is associated with improved OS (summarized HR = 0.42; 95%CI = 0.31–0.56) and PFS (summarized HR = 0.69; 95% CI = 0.45–1.07) regardless of diabetes status [117]. Several others retrospective studies reached similar conclusions, indicating that cancer patients with diabetes treated with metformin presented a substantially lower cancer burden than patients with diabetes treated with other antilipidemic drugs [118,119,120,121,122]. In vivo studies demonstrate that metformin can target ovarian cancer stem cells (CSCs) and, therefore, enhance chemotherapy response [103,123] proposing a mechanism of action based on CSC targeting [92,103,124,125,126,127,128].

A recent phase II clinical trial show a median PFS of 18.0 months (95% CI = 14.0–21.6) with a relapse-free survival at 18 months of 59.3% (95% CI = 38.6–70.5) and a median OS of 57.9 months (95% CI = 28.0-not estimable) for metformin users in non-diabetic OC patients [129]. The same authors demonstrate that metformin is a CSC targeting agent, since metformin-treated ex vivo tumours exhibit an average 2.4-fold decrease in ALDH^+^/CD133^+^ cells and an increased sensitivity to cisplatin compared to non-metformin-treated cells [129].

A phase I clinical trial is currently testing the effect of metformin in combination with carboplatin-paclitaxel chemotherapy in advanced OC and evaluating its safety and pharmacokinetics [130]. Additionally, a randomized phase-II trial is being conducted to evaluate metformin as a maintenance therapy in combination with standard chemotherapy in stage III–IV ovarian, fallopian tube or primary peritoneal cancer patients (NCT02122185).

Ongoing clinical trial are currently testing this therapeutic option the OC setting being determinant to unveil the role of metformin in the OC treatment (Appendix A). Many experimental and epidemiological results emphasize the utility of metformin in oncology management, suggesting its potential role in the treatment of advanced OC, especially when combined with platinum compounds.

### 2.3. Bisphosphonates

Bisphosphonates (e.g., alendronate and zoledronic acid) are widely used to treat osteoporosis through inhibition of bone resorption by decreasing osteoclast activity [131,132]. Bisphosphonates blocks farnesyl pyrophosphate synthase located downstream than HMGCR in the mevalonate pathway [133]. Farnesyl pyrophosphate is an enzyme involved in the synthesis of compounds that are necessary for maintaining osteoclast function, such as farnesyl pyrophosphate and geranylgeranyl pyrophosphate [134]. Bisphosphonates have an effect in prevention of specific clinical complications of bone metastasis (hypercalcemia, bone fractures, and pain) and have an additional anti-metastatic and anti-tumoral property when combined with chemotherapeutic agents inhibiting tumour proliferation and dissemination [134,135,136,137,138]. In vitro experiments, using breast and prostate cancer cell lines, demonstrate that bisphosphonates promote apoptosis and block cell proliferation [139,140]. In addition, studies using OC cell lines and animal models have shown that bisphosphonates have an anti-proliferative and pro-apoptotic activity through the inhibition of cell proliferation and angiogenesis, induction of apoptosis, and activation of immune cells [141,142,143,144,145,146]. A recent study demonstrated that bisphosphonates inhibit OC cell lines proliferation in a concentration-dependent manner in vitro and potentiate a delay in tumour formation and a decrease in tumour cell proliferation in transgenic OC mouse models (Figure 1) [147]. Studies showed that alendronate, a drug of the bisphosphonates class, reduce stromal invasion, tumour burden, and ascites, suggesting that has anti-tumoral effect in OC [148]. On the other hand, zoledronic acid have an anti-proliferative and anti-invasive activity and presents an interesting potential to delay recurrences in ovarian, endometrial, and breast cancers since, it disturbs relevant steps of tumour dissemination (i.e., invasion and colony formation) [134].

Some reports indicated that bisphosphonates exhibit anti-tumoral properties in many types of cancers, for example, they can inhibit the onset of breast (relative risk (RR) = 0.87; 95% CI = 0.81–0.93) [149] and endometrial (RR = 0.75; 95% CI = 0.60–0.94) [150] cancers. Other epidemiological study reported that bisphosphonates, administrated with or without statins, are associated with a reduced risk of OC [151]. In addition, it was reported that patients bisphosphonates users present a significantly reduced risk for ovarian (OR = 0.49; 95% CI = 0.26–0.93) and endometrial (OR = 0.39; 95% CI = 0.24–0.63) cancers compared to non-users [151].

Hence, it has been shown that bisphosphonates exhibit an anti-tumour effect and a capacity to delay recurrences in the OC setting; however, further studies are needed to evaluate its value as cancer treatment option.

### 2.4. Ivermectin

Ivermectin, a polycyclic lactone pesticide produced by S*treptomyces avermitilis* bacterium, is a broad-spectrum antiparasitic agent [152] that binds with high affinity to the glutamic acid operative chloride ion channel localized in nerve and muscle cells in invertebrates [153,154]. This ligation causes increased permeability in the cell membrane to enable chloride ions to enter the cells, resulting in hyperpolarization of nerve and muscle cells, which cause parasite paralysis and extinction [153,154].

Ivermectin is widely used to treat onchocerciasis, lymphatic filariasis, strongyloidiasis, scabies and head lice [155]. In addition, ivermectin can also exhibits other therapeutic actions such as, antibacterial, antiviral and anticancer [156]. As anti-neoplastic agent, it has been demonstrated that ivermectin exhibit anti-tumoral activity in different types of cancers, with emphasis on ovarian, colon and lung cancers, glioma, leukaemia and melanoma [157,158,159,160,161]. Different mechanisms can explain this activity, namely the inhibition of multidrug resistance proteins (MDR), modulation of protein kinase B (Akt)/mTOR and Wnt/T-cell factor (Wnt/TCF) signalling pathways, degradation of p21–activated kinase (PAK-1) and downregulation of stemness genes to preferentially target CSCs populations [158,159,161,162,163,164,165,166,167].

A recent in vitro and in vivo study demonstrated that ivermectin induce cytostatic autophagy in breast cancer cells linked to the inhibition of PAK-1 expression leading to a reduced phosphorylation of Akt and blockage of Akt/mTOR signalling pathway inhibiting tumour growth [163,168]. PAK-1 is abnormally expressed in various neoplasms, including breast, ovarian, pancreatic, colon and prostate cancers and is involved in tumour cell growth and development of chemoresistance [163,166]. Hashimoto et al. demonstrated in vitro that ivermectin induces inactivation of PAK-1 inhibiting OC cells growth [158]. Another recent study demonstrated that ivermectin preferentially targets breast CSCs by increasing the level of intracellular reactive oxygen species (ROS) associated with oxidative stress and DNA damage [161]. In addition, it has been shown that ivermectin could decrease multidrug resistance in breast cancer and enhance the cytotoxicity of doxorubicin and paclitaxel [169]. In glioblastoma cells, ivermectin inhibits angiogenesis and deactivates Akt/mTOR signalling pathway [166]. Moreover, ivermectin significantly inhibits proliferation and induces apoptosis in multiple renal cell carcinoma in vitro, and significantly delays tumour growth in vivo, by induction of mitochondrial dysfunction and oxidative stress [170]. Since ivermectin is a compound that targets yes-associated protein 1 (YAP1) [171], it is anticipated to exhibit anti-tumoral effects against ovarian, gastric, colorectal, and lung cancers, for which high expression of YAP1 is thought to be a prognostic indicator [172,173,174,175,176]. Ivermectin suppress the proliferation of gastric cancer cells in vitro and in vivo via YAP1 expression inhibition in a concentration and time-dependent manner [177]. In chronic myeloid leukaemia, it was demonstrated that ivermectin selectively induces apoptosis by mitochondrial dysfunction and oxidative stress induction [178].

In OC, in vitro and in vivo results show that ivermectin has a karyopherin subunit beta 1 (KPNB1) dependent anti-tumoral effect and the combination of ivermectin and paclitaxel produces a synergistic effect than each drug alone [179]. OC patients with high expression of KPNB1 present poor survival, consequently, ivermectin represents a promising candidate for combinatory treatment in OC. Recently, an interesting study suggest that ivermectin may useful OC combinatory treatments, demonstrating that this drug significantly augmented cisplatin inhibitory effect by suppressing the phosphorylation of key molecules in Akt/mTOR signalling pathway [180]. In same study, using an OC xenograft mouse models, authors showed that ivermectin alone inhibit tumour growth and, in combination with cisplatin, completely reversed tumour growth [180]. Corroborating these findings, another in vivo and vitro study demonstrated that ivermectin reverse the chemoresistance in colorectal, breast, and chronic myeloid leukaemia cancer cells by inhibit epidermal growth factor receptor (EGFR)/extracellular signal-regulated kinases (ERK)/Akt/nuclear factor kappa B (NF-κB) pathway (Figure 1) [181].

Previous findings demonstrate that ivermectin enhances the anti-cancer efficacy of chemotherapeutic drugs, especially in chemoresistant cells. Thus, further studies with ivermectin in combination with chemotherapeutic agents should be performed to validate its use in OC treatment.

### 2.5. Itraconazole

Itraconazole is a broad-spectrum antifungal agent that inhibits synthesis of ergosterol in the fungal cell membrane by blocking enzyme lanosterol 14a-demethylase, resulting in the destruction of the fungal membrane [182,183,184]. Itraconazole is widely used to treat fungal infections, including aspergillosis, candidiasis, and histoplasmosis, and in immunosuppressive disorders prophylaxis [184,185]. In addition, several experimental and clinical data show promising results regarding itraconazole antiangiogenic activity and has been repurposed as an anti-cancer agent in several types of cancers [184,186,187].

Itraconazole targets different oncobiology mechanisms, including reversing chemoresistance mediated by P-glycoprotein, inhibiting Hedgehog, mTOR, and Wnt/β-catenin signalling pathways, inhibiting angiogenesis and lymphangiogenesis [186,188,189,190,191,192,193,194,195]. A recent study demonstrated that itraconazole is capable of inhibiting mTOR signalling through different upstream mechanisms, e.g., AMPK activation and cholesterol trafficking inhibition that enhances an anti-proliferative and anti-angiogenic effect leading to an increased drug efficacy and reversion of chemoresistance [190]. In addition, an in vitro study showed that itraconazole has anticancer effects on oral squamous cell carcinoma through proteins expression downregulation of Hedgehog pathway by inhibiting cell proliferation and migration [196]. Another in vitro study in pancreatic cancer demonstrated that itraconazole inhibit viability, induce apoptosis and suppress invasion and migration by impaired transforming growth factor beta (TGF-β)/mothers against decapentaplegic homolog 2/3 (SMAD2/3) signalling [197]. In oesophageal cancer, it was demonstrated that itraconazole inhibits cell growth through activating AMPK signalling [198,199].

In the OC setting, a recent study showed a synergistic effect of combining itraconazole and paclitaxel to enhance efficacy in xenograft and patient-derived xenografts (PDX) models derived from OC chemoresistant patients [200]. The authors demonstrated that itraconazole antiangiogenic activity is attributable to its ability to inhibit the vascular endothelial growth factor receptor 2 (VEGFR2) and phosphorylation of ERK, hedgehog, and mTOR pathways (Figure 1) [200]. This study suggests that combining itraconazole and paclitaxel could enhance chemotherapeutic response in epithelial OC patients [200].

Several studies demonstrate that the treatment with itraconazole, in combination with other therapeutic agents, is effective in several types of cancers to increasing the drug efficacy and to overcome chemoresistance [201,202,203,204,205]. In fact, it has been shown that itraconazole can be an effective therapeutic agent for ovarian, breast, prostate, basal cell, non-small-cell lung, endometrial, gastrointestinal, bladder, and pancreatic cancers [186,187,188,202,203,205,206,207,208,209,210,211]. In pancreatic and in non-small-cell lung cancers it has been reported a prolonged survival when itraconazole is administrated as second-line therapy [202,208]. In OC patients unresponsive to platinum agents, the administration of itraconazole combined with taxane-based chemotherapy significantly improved PFS (HR = 0.24; *p* = 0.002) and overall survival (HR = 0.27; *p* = 0.006) [203]. Some results demonstrate that itraconazole treatment is beneficial to treat refractory malignancies, including ovarian clear cell, triple-negative breast, pancreatic, and biliary tract cancers [202,203,209,212]. In addition, some preclinical and clinical trial data indicate that itraconazole is capable of reversing the paclitaxel chemoresistance [213,214,215].

Currently, some clinical trials are evaluating the effect of itraconazole as a cancer therapeutic in non-small cell lung cancer (NCT02357836), oesophageal cancer (NCT02749513), basal cell carcinoma (NCT02120677), and OC (Appendix A).

Available preclinical and clinical trial data indicate that itraconazole can reverse the paclitaxel chemoresistance. Considering that OC patients frequently present taxane-chemoresistant recurrences, more experimental studies using OC cells are needed to further evaluate this antineoplastic property of this antifungal drug.

### 2.6. Ritonavir

Ritonavir is a protease inhibitors currently approved for human immunodeficiency virus (HIV) infection treatment to control acquired immune deficiency syndrome (AIDS)) [216]. Highly active anti-retroviral therapy (HAART) is the combination therapy (i.e., reverse transcriptase inhibitors, such as, zidovudine and protease inhibitors, such as, ritonavir and nelfinavir) used to treat patients with HIV infection [217] characterized by an increased risk to develop several types of tumours [217,218,219]. Interestingly, some studies demonstrated that, in HIV patients treated with HAART [220,221,222,223], the incidence rate of some cancers is, in fact, lower. Corroborating these findings, results from other studies reported associations of a potential anti-neoplastic impact of HAART [224,225,226,227]. It has been shown that ritonavir induces apoptosis of lymphoblastic tumour cells including lymphoma, myeloid leukaemia, fibrosarcoma, mastocytoma, and Kaposi’s sarcoma [228,229,230,231]. An in vitro study demonstrated that ritonavir has an effective anti-proliferating activity in OC cells, inducing cell cycle arrest and apoptosis by inhibiting AKT pathway and retinoblastoma phosphorylation [217]. Ritonavir decreases the amount of phosphorylated AKT in a dose-dependent manner, which inhibits the phosphatidylinositol 3-kinase (PI3K)/Akt pathway having anti-tumoral effect (Figure 1). Further results in OC, demonstrate that ritonavir inhibit invasion and migration and has an additive effect when combined with paclitaxel treatment [217].

The evidences on ritonavir action on cancer cell are limited and need further validation. However, these existing data demonstrate that ritonavir has a potential for OC treatment in combination with conventional chemotherapy.

## 3. Using Ex Vivo Models to Test Individual Drug Repurposing Efficacy

The development of patient-derived cancer cells models to perform drug efficacy tests capable of predicting drug responses constitutes an important step towards a personalized cancer treatment. These tests should consider the phenotype, genotype and protein expression background of each patient tumour to search for effective drugs in accordance to an individual profile of drug response. Determining the chemoresistance profile in each individual patient could facilitate the discovery of the most effective drug for repositioning and treatment could be adapted accordingly [232]. Several ex vivo model systems have been exploited for these purposes: (a) 2D monolayer culture of dissociated cells, including primary cultures and cell lines; (b) 3D tumour cultures, comprising cell lines and organoids; (c) PDX; and (d) organotypic tumour tissue slices. The advantages and disadvantages of each of these systems were reviewed elsewhere [233]. Within these, patient-derived cancer cell culture systems standout, being an easily achievable, non-animal, preclinical model for drug efficacy tests capable of guiding clinical decisions [234]. Indeed, 2D and 3D models may be used for high-throughput drug sensitivity screening, which have started to show promising results with significant higher response rates in patients following guided treatments [235]. Specifically, in epithelial OC, ex vivo models have been shown to faithfully recapitulate phenotypic and genotypic tumour features [236]. Furthermore, results of drug profiling using these OC models showed sensitivity to drugs, such as, platinum and iPARPs, which correlated with patient clinical responses [237]. Likewise, ex vivo models are also suitable for testing high-throughput drug repurposing, as demonstrated in other cancers [238]. In epithelial OC, systematic drug repurposing based on clues from ex vivo testing of human OC patient cells has also began to be explored with case reports of patients showing response to drugs not usually applied [239].

Patient-derived cancer cell models are most commonly established from primary tumours, but can also be derived from metastases, and circulating tumour cells from blood, effusions, or other organic fluids [240]. The choice of tumour samples to be used for drug testing may be crucial to accurately predict patient response over the course of disease with impact in survival. In this respect, epithelial OC constitutes a particular setting given that its primary route of progression is in the peritoneal cavity [241]. Exfoliated tumour cells from the ovary or Fallopian tube are carried by physiological peritoneal fluid and disseminate throughout the abdominal cavity. Extensive peritoneal seeding by tumour cells is often associated with ascites, which may be, detected at the time of diagnosis and more frequently in advanced stages [242]. Malignant ascites comprises a set of cytokines, growth factors, mesothelial cells, macrophages, lymphocytes and tumour cells [243]. The easy accessibility of the peritoneal cavity makes ascites a powerful source of tumour cells, suitable for detecting prognostic and predictive biomarkers, as well as to perform ex vivo assays that could boost the discovery of personalized therapeutic approaches. In the absence of ascites, at diagnosis, tumour tissue for ex-vivo models can be collected simultaneously with sampling for pathology diagnostic purposes. In the setting of recurrence without ascites, biopsy for ex-vivo models should be carefully considered by clinicians together with their patients, as this is an invasive procedure. Establishment of cultures from blood circulating tumour cells, even though challenging, may be an alternative.

The establishment of ex vivo models from ascitic fluid-derived cancer cells (Figure 2) has been explored as a model to overcome some limitations of strongly manipulated cell lines and potentiate a proof-of-concept for personalized drug efficacy testing [232,234,244,245,246]. One limitation of drug repurposing is the low potency of hit compounds as single agents, as their maximal tolerated dose is often sub therapeutic for cancer treatment purposes [247]. Wisely, combined cocktails using chemotherapeutic drugs and one or more non-oncology drugs can be tested in ascitic fluid-derived cancer cells to deliver an individualized therapeutic solution to OC patients. The drug-repurposing concept aligned to this personalized testing system could further validate the existing evidences supporting the use of these old drugs for cancer treatment.

## 4. Conclusions

Several researchers worldwide are already investigating recycling the purpose of drugs not originally designed for cancer treatment. These therapies can play an important role in cancer chemoprevention, in recurrence delay, and as partners of old and new therapies, such as chemotherapy. However, this therapeutic strategy faces many obstacles concerning competition with new drug development and patent-related considerations. Many of these difficulties led to the lack of investment by the pharmaceutical industry that impairs the development of research projects that could further validate the current scientific evidences. Drug repurposing will ultimately benefit patients and low-income countries with economical fragile healthcare systems and limited access to new expensive drugs. Thus, off-patent drug research should have a fair financial support to guarantee an accurate result comparison from new and old drugs in cancer settings. Regarding OC, tumour cells present in the ascitic fluid constitute an opportunity to study ex vivo the efficacy of drugs in a personalized manner. This model has great potential to predict the synergistic effect between chemotherapy and repurposing drugs, leading to the best benefit of cancer patients.

## Figures and Tables

**Figure 1 ijms-21-07768-f001:**
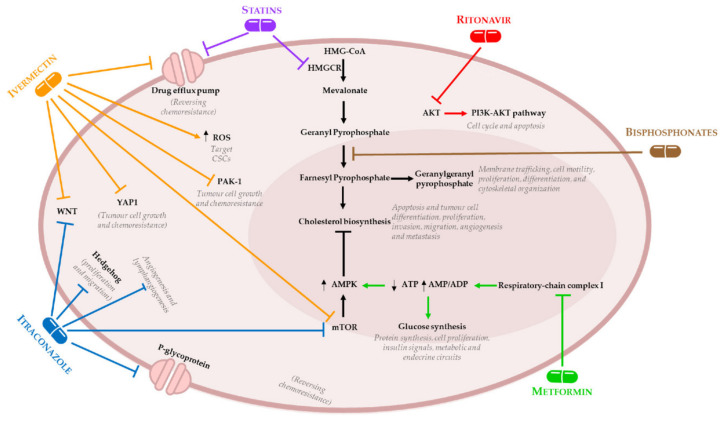
Mechanism of action of non-oncological drugs in ovarian cancer (OC). Statins inhibit 3-hydroxy-3-methylglutaryl-coenzyme A reductase (HMGCR) leading to the blocking of cholesterol biosynthetic pathway though a mevalonate-dependent mechanism. Moreover, statins can block drug efflux pumps by a mevalonate-independent mechanism. Bisphosphonates block farnesyl pyrophosphate synthase, located downstream HMGCR, leading to the impairment of cholesterol biosynthesis. Metformin inhibits insulin signals and glucose synthesis via respiratory-chain complex I blockage. Ritonavir is a protease inhibitor that inhibits the production of phosphorylated protein kinase B (AKT) leading to the impairment of phosphatidylinositol 3-kinases (PI3K)-Akt pathway. Itraconazole can inhibit Hedgehog, mammalian target of rapamycin (mTOR), and Wnt signalling pathway. Moreover, itraconazole can inhibit angiogenesis and lymphangiogenesis, and promote the overexpression of P-glycoprotein. Ivermectin interferes with several cellular mechanisms, including multidrug resistance proteins (MDR) inhibition, Akt/mTOR, and Wnt signalling pathways modulation, p21–activated kinase (PAK-1) and yes-associated protein 1 (YAP1). Moreover, ivermectin promotes the increase of intracellular reactive oxygen species (ROS) levels leading to the downregulation of stemness genes. Adenosine diphosphate (ADP); adenosine monophosphate (AMP); adenosine triphosphate (ATP); 3-Hydroxy-3-methylglutaryl-coenzyme A (HMG-CoA).

**Figure 2 ijms-21-07768-f002:**
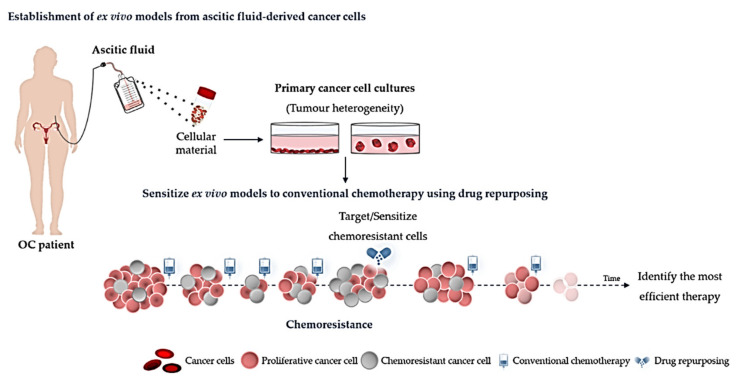
Establishment of ex vivo models from ascitic fluid-derived cancer cells to perform drug efficacy tests. Combination of drug repurposing (e.g., pitavastatin, metformin, bisphosphonates, ivermectin, itraconazole and ritonavir) with conventional chemotherapy (e.g., carboplatin and paclitaxel) may have the benefits of increased efficacy and has potential to decrease the risk of therapeutic failure. The effectiveness of drug repurposing approaches to target or sensitize chemoresistant cells to conventional chemotherapy can be validate in established ex vivo models. A schematic diagram demonstrating the conventional chemotherapy in combination with compounds of drug repurposing that can directly target the chemoresistant cell and tumour loses its ability to generate new cancer cells, or sensitize chemoresistant cells in order to disrupt the stemness and make them more sensitive to conventional chemotherapy.

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
