# Peer review of "Recycling the Purpose of Old Drugs to Treat Ovarian Cancer"

_ijms, 2020, doi:10.3390/ijms21207768_

Round 1

Reviewer 1 Report

The review by Nunes M et al.  'Recycling the purpose of old drugs to treat ovarian cancer' is a well written review that covers a substantial literature on the topic of interest.  The authors have put forward an elegant ascites model as a proof-of-concept to test personalized drug efficacy in patients.  Even though the concept is ideal but the fact remains that only 30-40% of ovarian cancer patients present with ascites at first diagnosis of cancer.  In case where ascites in not present in patients at diagnosis what alternative models should be considered?  Few lines along that thought would be helpful.  Alternatively, as ascites is present in most recurrent patients this unique model would be best to test old drugs in combination with chemotherapy to circumvent recurrence in patients.

Minor comments:

There are few typos which should be removed.

Page 3, line 137, it should be epithelial-mesenchymal transition (EMT) not the other way round.

Page 4, line 163, describe CSCs before abbreviation.

Page 5, line 249 describe KPNB1 before abbreviation.

Author Response

The Reviewer recognizes the interest of the Ms and highlights a relevant percentage of ovarian cancer patients that do not present ascites at diagnosis and, therefore, ex-vivo cultures based on ascitic fluid cannot be applied. We completely agree with the Reviewer and we recognize this may be a major limitation. At diagnosis, all patients undergo tissue collection for pathological examination and establishment of diagnosis. Thus at this time, if ascitis is not present, we can also collect tissue for ex-vivo cultures concomitant to tissue for diagnostic purposes. The major limitation is in the setting of tumour recurrence that does not associate with ascites, where, the alternative could be to perform a small biopsy to retrieve cancer cell for ex-vivo cultures. However, as this is an invasive procedure, it should be carefully evaluated by clinicians in collaboration with their patients. Additionally, as we state in the Ms, patient derived cancer cell models can also be established from circulating tumour cells from blood.

Following the reviewer suggestions, we added a few lines, emphasizing this issue (page 9, line 415):

“In the absence of ascites, at diagnosis, tumour tissue for ex-vivo models can be collected simultaneously with sampling for pathology diagnostic purposes. In the setting of recurrence without ascites, biopsy for ex-vivo models should be carefully considered by clinicians together with their patients, as this is an invasive procedure. Establishment of cultures from blood circulating tumour cells, even though challenging, may be an alternative.”

Reviewer 2 Report

The review article, "Recycling the purpose of old drugs to treat ovarian cancer", provides comprehensive review of alternative/repurposing of drugs already in use. The authors have done very thorough review of marketed drugs which may be potentially useful for Ovarian Cancer. Authors have also provided in vitro and ex vivo approaches to assess therapeutic potential of these drugs in case of ovarian cancer. The manuscript is well crafted and there are minor suggestion which will boost this manuscript a notch. 

Minor Comments: 

  1. in line 21, choose different word instead of 'important'. Important side effect is not a good choice. Just side effects or potential side effect or anything appropriate will work better. 
  2. Please provide median progression free survival % or rate or years in line 44
  3. In line 45-46, choice of word 'concerning' is not good. Please change. 
  4. IN line 102, please change "These results' to These Studies... The results are not generated by authors to support this review, Authors are using these studies to support there conclusion in this review so it should be written that way. 
  5. Line 110, half-time life is incorrect. Does author meant half-life..
  6. In line, 113, should it be 'anti-hyperglycemia'?

Major Comment: 

  1. Authors should provide figures/scheme for each drugs and there possible mechanism as anti-proliferative effect proposed in Ovarian Cancer. This is very important for reader engagement and quick understanding of the anti-tumor Mechanism of action of the drugs. 

Author Response

The Reviewer recognizes the scientific soundness of the Ms, identifies some minor corrections and suggest the incorporation of a new figure to further elucidate the mechanism of action of the repurposing drugs described in this Ms.

Minor comments:

  1. The text was corrected in the Ms (track changes: Page 1, line 21)
  2. We included a new sentence with more detailed information on patient survival (track changes: Page 2, lines 45 and 46)
  3. The text was corrected in the Ms (track changes: Page 2, line 48)
  4. The text was corrected in the Ms (track changes: Page 3, line 142)
  5. The text was corrected in the Ms (track changes: Page 4, line 150)
  6. The text was corrected in the Ms (track changes: Page 4, lines 153 and 154)

Major comments:

1. Acknowledging the importance of this suggestion, we included a new figure that compiles the main mechanisms of action the therapeutic drugs mentioned in the Ms (Figure 1, Page 2, lines 62-97)

Reviewer 3 Report

The article presents a model for the use of old drugs, already well known, for the treatment of ovarian cancer. Despite a very interesting topic, the article does not meet the requirement of scientific innovation, and the proposed model has already been tested many times in several centers around the world. Despite the correct description of the mode of action of the drugs, the authors did not analyze all clinical trials in the world testing drugs for ovarian cancer. And the presented method of testing drugs is also already known and has been used in other clinical trials.

Author Response

The reviewer acknowledges that drug repurposing is a very interesting topic and recognizes that the description of the mechanism of action of the drugs is correct. However, the reviewer calls into questions the scientific novelty of this Review.

The main aim of this review was to strengthen the importance of repurposed drugs as therapeutic alternatives in the ovarian cancer setting and propose an in vitro model to assess its therapeutic value in a personalized manner. We recognize that the ex vivo model is already in use but mainly applied in large scale drug screening tests, and are not extensively explored to evaluate the efficacy of non-oncological drugs such as the presented in the Ms. Here, we propose a personalized test that could add new important data on the possibility to include these drugs to each patient therapeutic scheme.

Round 2

Reviewer 3 Report

Accept.